

# Non-standard neutrino spectra from annihilating neutralino dark matter

**Melissa van Beekveld[1*], Wim Beenakker[2,3†], Sascha Caron[2,4‡],
Jochem Kip[2§], Roberto Ruiz de Austri[5°] and Zhongyi Zhang[2,4△]**

**1** Rudolf Peierls Centre for Theoretical Physics, Clarendon Laboratory,
Parks Road, University of Oxford, Oxford OX1 3PU, UK
**2** Institute for Mathematics, Astrophysics and Particle Physics,
Radboud University Nijmegen, Heyendaalseweg 135, Nijmegen, the Netherlands
**3** Institute of Physics, University of Amsterdam,
Science Park 904, 1018 XE Amsterdam, The Netherlands
**4** Nikhef, Science Park, Amsterdam, The Netherlands
**5** Instituto de Física Corpuscular, CSIC-Universitat de València,
E-46980 Paterna, Valencia, Spain

⋆ melissa.vanbeekveld@physics.ox.ac.uk , † w.beenakker@science.ru.nl , ‡ scaron@nikhef.nl ,
§ jochem.kip@ru.nl , ∘ rruiz@ific.uv.es , △ zzhang@nikhef.nl

## Abstract

Neutrino telescope experiments are rapidly becoming more competitive in indirect detection searches for dark matter. Neutrino signals arising from dark matter annihilations are typically assumed to originate from the hadronisation and decay of Standard Model particles. Here we showcase a supersymmetric model, the BLSSMIS, that can simultaneously obey current experimental limits while still providing a potentially observable non-standard neutrino spectrum from dark matter annihilation.

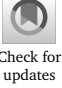

# 1 Introduction

One of the big unsolved mysteries in current-day physics is the exact nature of dark matter (DM), which as described by the Lambda-CDM model should make up 85% of the matter content of the universe [1]. A standard hypothesis is that DM is comprised of Weakly Interacting Massive Particles (WIMPs), which for example can naturally be provided by extensions of the Standard Model (SM) in which supersymmetry (SUSY) is imposed and *R*-parity conservation is assumed. Collider, direct, and indirect-detection experiments have not yet found any conclusive evidence for the existence of WIMPs. While a number of different studies have found excesses that may indicate the presence of DM, for example the AMS-02 antiproton over proton ratio [2–9], and the Fermi-LAT gamma-ray data [10–18], these excesses remain small and subject to large uncertainties regarding production and propagation [19–27]. An attractive messenger particle that circumvents these uncertainties is the neutrino, as these particles can propagate freely through the Galaxy, thus providing a potentially clean signal due to their low interaction rate. This low interaction rate is of course also challenging for detection purposes; it is only in recent years that cosmic-neutrino detection experiments have become potentially sensitive to neutrinos originating from DM annihilation [28–30]. These spectra typically arise from SM final-state particles, e.g. $b\bar{b}$, $\tau^+\tau^-$, or $\nu_{e,\mu,\tau}\nu_{e,\mu,\tau}$, in which $\nu_{e,\mu,\tau}\nu_{e,\mu,\tau}$ provide the most stringent bounds due to the monochromatic neutrino line [31]. There have been studies showing that, at least in the Minimally Supersymmetric Standel Model (MSSM), the neutrino spectra can deviate somewhat from only those resulting from SM final-state particles [32]. The construction of KM3NeT is expected to increase this sensitivity, especially since neutrino signals arising from the Galactic Centre could be measured with unprecedented precision, thus providing an opportune way of investigating DM via neutrino physics.

The SM, defined with only left-handed neutrinos, and its minimal supersymmetric extension, the MSSM, both lack a mechanism for generating neutrino masses. One example of a model that incorporate neutrino masses (and a DM sector) is the *B-L*-extended supersymmetric SM with inverse seesaw (BLSSMIS) [33–35]. The inverse seesaw provides a mechanism for naturally light neutrinos by introducing right-handed neutrinos and an additional neutrino field with a small mass term. In the SM *B-L* (baryon number minus lepton number) is an accidental symmetry and is always conserved, as opposed to *B* and *L* symmetry separately, which are for example violated in sphaleron processes [36]. This apparent accidental symmetry can be generated naturally from a $U(1)_{B-L}$ gauge group. Neutrino masses in the BLSSMIS are generated via an inverse-seesaw mechanism [37–42]. This allows us to study the effects of a non-minimal neutrino mass model on the spectrum of cosmic neutrinos originating from DM annihilation.

This paper is structured as follows. The relevant parameters, particle content, and other details of the BLSSMIS are discussed in section 2. The scanning methodology and the cuts that are imposed upon all data points is shown in section 3. Various DM observables are discussed in section 4, starting with the relic density, before moving to the current direct-detection limits and then discussing the LHC limits, and ending with a more in-depth study into the neutrino spectra of annihilating neutralinos. Lastly, in section 5, the concluding remarks are made.

## 2 The BLSSMIS model

### 2.1 Particle content

The total particle content of the BLSSMIS is largely the same as the MSSM, with the difference being a new $Z'$ boson arising from the added $U(1)_{B-L}$ gauge group, and additional particles in the neutrino, sneutrino, higgs, and neutralino sectors. There are nine majorana neutrino mass eigenstates in this model due to the inverse seesaw mechanism, of which three are light SM-like neutrinos and six are heavy neutrinos. Naturally, this also extends the sneutrino sector, which contains eighteen real scalar fields. Two new higgs singlets are added to provide a mass to the $Z'$ boson, so consequently three new higgs mass-eigenstates (two CP-even and one CP-odd) appear additionally to those of the MSSM. The neutralinos are extended from four in the MSSM to seven neutralinos in the BLSSMIS; one new bino-like field and two higgsino-like fields.

### 2.2 Superpotential and Lagrangian

The increased particle content of the BLSSMIS leads to additional fields in the superpotential. The two new higgs singlet fields are denoted by $\eta$ and $\overline{\eta}$. Furthermore, to implement the inverse seesaw mechanism, six fermion singlet fields are added: three right-handed neutrinos, and three fields that are only charged under $U(1)_{B-L}$, collectively called $s_2$. To make the theory anomaly free, three additional singlet fields are required, denoted by $s_1$, which cancel the anomaly introduced by adding the $s_2$ fields. The $s_1$ fields will be integrated out. Naturally, all fields are implemented as superfields since this model is supersymmetric. The resulting superpotential then follows as

$$W = \mu \hat{H}_u \hat{H}_d - \mu_\eta \hat{\eta}\hat{\overline{\eta}} + \hat{U}^c Y_u \hat{Q}\hat{H}_u - \hat{D}^c Y_d \hat{Q}\hat{H}_d + \hat{E}^c Y_e \hat{L}\hat{H}_d - \hat{\nu}^c Y_\nu \hat{L}\hat{H}_u + \hat{\nu}^c Y_x \hat{\eta}\hat{s}_2 + \hat{s}_2 \mu_S \hat{s}_2 \,. \tag{1}$$

Here $\hat{Q}$, $\hat{L}$ are the left-handed quark and lepton superfields respectively, $\hat{U}$, $\hat{D}$, $\hat{E}$, $\hat{\nu}$ are the right-handed up-type quark, down-type quark, lepton, and neutrino superfields and $\hat{s}_2$ is the superfield belonging to $s_2$. Furthermore, $\hat{H}_u$, $\hat{H}_d$ denote the up-type and down-type higgs superfields, and $\hat{\eta}$, $\hat{\overline{\eta}}$ are the aforementioned two new higgs singlet superfields. The Yukawa $3 \times 3$ matrices are indicated with $Y_i$ where the subscript $i$ indicates the corresponding fields. We break SUSY by assuming Super Gravity (SUGRA) with unification at the Grand Unified Theory (GUT) scale, except for the gaugino breaking parameters. We define the GUT scale as the energy at which the coupling constants of the gauge groups all have a single unified value. The breaking terms at the SUSY scale, whose value here is defined as the geometric mean of the two stop masses, are obtained via the evolution of the renormalisation group equations (RGEs). The resulting Lagrangian containing the soft-SUSY breaking terms then reads

$$
\begin{aligned}
-\mathcal{L}_{\text{break}} =\ & \tilde{q}_L^* m_{\tilde{q}}^2 \tilde{q}_L + \tilde{l}_L^* m_{\tilde{l}}^2 \tilde{l}_L + \tilde{e}_R^* m_{\tilde{e}}^2 \tilde{e}_R + \tilde{\nu}_R^* m_{\tilde{\nu}}^2 \tilde{\nu}_R + \tilde{u}_R^* m_{\tilde{u}}^2 \tilde{u}_R + \tilde{d}_R^* m_{\tilde{d}}^2 \tilde{d}_R + \tilde{s}_2^* m_{\tilde{s}_2}^2 \tilde{s}_2 \\
&+ (\tilde{d}_R^* T_d \tilde{q}_L H_d + \tilde{e}_R^* T_e \tilde{l}_L H_d + \tilde{\nu}_R^* T_\nu \tilde{l}_L H_u + \tilde{u}_R^* T_u \tilde{q}_L H_u + \eta \tilde{\nu}_R^* T_x \tilde{s}_2 + \text{h.c.}) \\
&+ m_{H_u}^2 |H_u|^2 + m_{H_d}^2 |H_d|^2 + m_\eta^2 |\eta|^2 + m_{\overline{\eta}}^2 |\overline{\eta}|^2 + B_\mu H_d H_u + B_\eta \eta \overline{\eta} \\
&+ \frac{1}{2} M_1 \tilde{B}\tilde{B} + \frac{1}{2} M_2 \tilde{W}\tilde{W} + \frac{1}{2} M_3 \tilde{G}\tilde{G} + \frac{1}{2} M_{BL} \tilde{B}_{BL}\tilde{B}_{BL} + M_{BBL} \tilde{B}\tilde{B}_{BL} + \text{h.c.} \tag{2}
\end{aligned}
$$

The product between two doublets is defined as a contraction with the antisymmetric tensor $\epsilon^{ab}$. The $m_i^2$ are the sfermion and higgs mass terms, in which $i$ again indicates the relevant field. The sfermion mass terms are matrices while the higgs mass terms are simply scalars. These scalar-breaking terms all have a common value $m_0$ at the GUT-scale. The $T_i$ are trilinear couplings, defined as $Y_i A_0$, in which $Y_i$ is the Yukawa coupling and $A_0$ a common GUT scale

term. The gaugino mass parameters $M_1$, $M_2$, $M_3$, $M_{BL}$ and $M_{BBL}$ are usually defined via a common $M_{\frac{1}{2}}$ at the GUT scale, but in this work each gaugino mass has its own GUT-scale parameter, i.e. $M_1$, $M_2$, $M_3$, $M_{BL}$ for the $\widetilde{B}$, $\widetilde{W}$, $\widetilde{G}$, and $\widetilde{B}_{BL}$ respectively. The mixing term between $\widetilde{B}$ and $\widetilde{B}_{BL}$, $M_{BBL}$, is set to zero at the GUT-scale. The splitting of the gaugino mass term grants greater freedom in the neutralino sector and is allowed in SUGRA models as the unification of the different gaugino mass parameters is not required, merely often implemented to reduce the number of parameters. In addition to the standard $\tan(\beta)$ parameter, which defines the value for the ratio of the two higgs vacuum expectation values (vevs) $v_u/v_d$, there is an additional $\tan(\beta')$ that represents the ratio of the two new vevs, $v_\eta/v_{\overline{\eta}}$. Furthermore, we fix $|\mu|^2$, $|\mu_\eta|^2$, $\mathfrak{Re}(B_\mu)$, and $\mathfrak{Re}(B_\eta)$ using the tadpole equations that follow from the minimisation of the higgs scalar potentials. Additionally, the charges under $U(1)_{B-L}$ are fixed for all fields except the new higgs fields $\hat{\eta}$ and $\hat{\overline{\eta}}$, and the new neutrino fields $\hat{s}_1$ and $\hat{s}_2$. We choose the charges of $\hat{s}_1$ and $\hat{s}_2$ to be 1/2 and -1/2 respectively under $U(1)_{B-L}$. This choice fixes the $U(1)_{B-L}$ charge of $\hat{\eta}$ and $\hat{\overline{\eta}}$ to be -1 and 1 respectively. A more detailed discussion on the charges of the superfields and the corresponding superpotentials can be found in appendix A. The foregoing implies that the coupling strength of the $Z'$ boson to all fields is fixed by the theory and cannot be scaled manually. The only impact that the model parameters have on interactions involving the $Z'$ particle is the mass of the $Z'$, and the mixing and mass of the mass eigenstates of the particles coupling to the $Z'$.

While at the GUT scale the couplings are unified, mixing between the $U(1)_Y$ and $U(1)_{B-L}$ occurs at the SUSY scale. This results in the gauge couplings in the covariant derivative appearing in the dynamic terms of the Lagrangian becoming a 2×2 matrix. These mixing effects are a priori expected to have an impact on the electroweak sector, but this can be removed by redefining the two gauge fields. The fields before mixing, $B'$ and $B'_{B-L}$, can be redefined into $B$ and $B_{B-L}$ such that the electroweak sector is left untouched. This is done by performing a simple rotation, i.e.

$$D^\mu = \partial^\mu + i \begin{pmatrix} Y_B & Y_{B-L} \end{pmatrix} \begin{pmatrix} g_Y & g_{YBL} \\ g_{BLY} & g_{BL} \end{pmatrix} \begin{pmatrix} B'^\mu \\ B'^\mu_{B-L} \end{pmatrix} \equiv \partial^\mu + i \begin{pmatrix} Y_B & Y_{B-L} \end{pmatrix} \begin{pmatrix} g_1 & g_\times \\ 0 & g_{B-L} \end{pmatrix} \begin{pmatrix} B^\mu \\ B^\mu_{B-L} \end{pmatrix}, \quad (3)$$

where

$$g_1 = \frac{g_Y g_{BL} - g_{YBL} g_{BLY}}{\sqrt{g_{BL}^2 + g_{BLY}^2}}, \qquad g_\times = \frac{g_Y g_{BLY} + g_{BL} g_{YBL}}{\sqrt{g_{BL}^2 + g_{BLY}^2}}, \qquad g_{B-L} = \sqrt{g_{BL}^2 + g_{BLY}^2}. \quad (4)$$

Here $g_1$ and $g_{B-L}$ are the $U(1)_Y$ and $U(1)_{B-L}$ coupling constants respectively. At the GUT scale, all coupling constants are unified, while the off-diagonal elements are zero. This completely fixes the new coupling constants and thus they cannot be tuned freely.

The mixing of the $U(1)$ gauge groups impacts the $M_{BLL}$ term, but also the neutralino sector as it introduces an explicit mixing term between the $\widetilde{B}_{BL}$ and the higgsino fields. In the basis of $\begin{pmatrix} \widetilde{B} & \widetilde{W} & \widetilde{h}_d & \widetilde{h}_u & \widetilde{B}_{BL} & \widetilde{h}_\eta & \widetilde{h}_{\overline{\eta}} \end{pmatrix}$, the mass matrix of the neutralinos is given by

$$M_{\widetilde{\chi}^0} = \begin{pmatrix} M_1 & 0 & -\frac{1}{2} g_1 v_d & \frac{1}{2} g_1 v_u & M_{BB'} & 0 & 0 \\ 0 & M_2 & \frac{1}{2} g_2 v_d & -\frac{1}{2} g_2 v_u & 0 & 0 & 0 \\ -\frac{1}{2} g_1 v_d & \frac{1}{2} g_2 v_d & 0 & -\mu & -\frac{1}{2} g_\times v_d & 0 & 0 \\ \frac{1}{2} g_1 v_u & -\frac{1}{2} g_2 v_u & -\mu & 0 & \frac{1}{2} g_\times v_u & 0 & 0 \\ M_{BB'} & 0 & -\frac{1}{2} g_\times v_d & \frac{1}{2} g_\times v_u & M_{BL} & -g_{B-L} v_\eta & g_{B-L} v_{\overline{\eta}} \\ 0 & 0 & 0 & 0 & -g_{B-L} v_\eta & 0 & -\mu_\eta \\ 0 & 0 & 0 & 0 & g_{B-L} v_{\overline{\eta}} & -\mu_\eta & 0 \end{pmatrix}. \quad (5)$$

Here the upper-left 4×4 matrix can be recognised as the MSSM neutralino mixing matrix. When refering to the gaugino content of the mass eigenstates, in what follows, we will combine the gauge eigenstates $\widetilde{h}_d$ and $\widetilde{h}_u$ into the total higgsino component $\widetilde{h}$, defined through

$\widetilde{h} = \sqrt{\widetilde{h}_u^2 + \widetilde{h}_d^2}$. The $\widetilde{h}_\eta$ and $\widetilde{h}_{\overline{\eta}}$ will similarly be combined into $\widetilde{h}_{BL}$. The neutralino mixing matrix is diagonalized as $N^* M_{\widetilde{\chi}^0} N^{-1} = M_{\widetilde{\chi}^0}^D$. The neutralino mass eigenstates are given by $\widetilde{\chi}_{1\ldots7}^0$, which are mass ordered. The chargino sector is the same as in the MSSM, and thus shall not be discussed here.

The mass matrix for the neutrinos can be read from the superpotential (1) and in a basis of $\begin{pmatrix} \nu_L & \nu_R & s_2 \end{pmatrix}$ it reads

$$M_\nu = \begin{pmatrix} 0 & \frac{1}{\sqrt{2}} v_u Y_\nu^T & 0 \\ \frac{1}{\sqrt{2}} v_u Y_\nu & 0 & \frac{1}{\sqrt{2}} v_\eta Y_x \\ 0 & \frac{1}{\sqrt{2}} v_\eta Y_x^T & \mu_S \end{pmatrix} \equiv \begin{pmatrix} 0 & M_\nu^T & 0 \\ M_\nu & 0 & M_X \\ 0 & M_X^T & \mu_S \end{pmatrix}. \tag{6}$$

Here $M_\nu = v_u Y_\nu / \sqrt{2}$ and $M_X = v_\eta Y_x / \sqrt{2}$. Since the $\mu_S$ parameter breaks the $B - L$ symmetry it can only receive a small non-zero value via high-scale radiative effects [38]. Assuming $\mu_S$ to be small automatically leads to three light neutrino mass eigenstates ($\nu_l$) and six heavy ($\nu_{h_{1,2}}$) mass eigenstates. Furthermore, taking $M_\nu$, $M_X$, and $\mu_S$ to be $m_\nu I$, $m_X I$ and $\mu_S I$ respectively, where $I$ is the $3 \times 3$ identity matrix, we find the tree-level neutrino mass eigenstates as

$$\nu_l \approx \frac{m_\nu^2}{m_\nu^2 + m_X^2} \mu_S, \qquad\qquad \nu_{h_1}^2 \approx \nu_{h_2}^2 \approx m_\nu^2 + m_x^2. \tag{7}$$

The mass eigenstates of the heavy neutrino have no suppression of the left-handed neutrino field, and thus couple without suppression to the higgs, $W^\pm$, and $Z$ boson. Additionally, a coupling to the $Z'$ via the $B\text{-}L$ charge of the left and right-handed neutrinos is present. Both aforementioned couplings result in the $\nu_h$ being an unstable particle. We shall refer to both $\nu_{h_1}$ and $\nu_{h_2}$ as $\nu_h$ for the remainder of the text, as they are phenomenologically equivalent for our purposes due to their similar mass, decay width and decay modes. Thus of the ab initio three DM candidates of the B-L-SSM-IS, i.e. the heavy neutrino, sneutrino, and neutralino, the heavy neutrino can be discarded by this assumption as a DM candidate. However, while excluded as a DM candidate, the heavy neutrinos play a significant role in the neutrino spectrum of annihilating neutralinos for which $\widetilde{B}_{BL}$ or $\widetilde{h}_{BL}$ is the largest component. Furthermore, in the following we shall only look at neutralino dark matter, as sneutrino dark matter in the B-L-SSM-IS has been sufficiently investigated in previous works where it is shown that it is indeed a viable DM candidate [33, 34].

## 3 Scanning methodology

The objective of our scanning procedure was to find solutions with novel phenomenology, and not to completely map out high-likelihood regions in the parameter space. In order to do this we implemented the following search strategy. Three different initial parameter ranges were used, which are given in table 1. We scanned these parameter spaces using a particle filter [17, 43–47] in which we applied hard cuts on the exclusion limits, which will be explained below, in order to increase the efficiency of the sampling. The scanning procedure was performed in three steps. As a first step, the entire parameter space was sampled randomly and uniformly, with the exception of $Y_\nu$, $Y_x$, and $\mu_S$ which are sampled logarithmically. The dataset that this yielded was used as the seed for the second step of the sampling procedure, in which a Gaussian particle filter was used to find regions of the parameter space that provide a DM relic density of 0.12 or less. Having found these viable regions, we manually identified the interesting regions, namely those with novel neutrino phenomenology, contained in this dataset and fed those as new input into a Gaussian particle filter in order to zoom in on those regions

of interest. In general we decreased the gaussian width by 0.5 per iteration, but this decrease is occasionally manually adjusted in order to keep a good resolution. We stopped the scan after we found solutions with new phenomenology. Note that $\mu_S$ is chosen to be small, in accordance with its nature as discussed in the previous section. Furthermore, we choose the positive branch of both sign($\mu$) and sign($\mu_\eta$). The three ranges denoted in table 1 are used to investigate the high-mass parameter space, and no significant difference was found between the three ranges, aside from the presence of more high-mass solutions when the parameter ranges are increased. In total $\mathcal{O}(10^7)$ points have been considered.

Sarah 4.14.3 [48–53] is used as input for SPheno-4.0.4 [54, 55], which is employed as

Table 1: The initial parameter ranges for the three different scanning ranges. The parameters are defined and sampled at the GUT scale.

| # | $m_0$  $M_{1,2,3,BL}$ (GeV) | $A_0$ (GeV) | $\tan(\beta)$ | $\tan(\beta')$ | $Y_\nu$  $Y_x$ | $\mu_S$ (GeV) |
|---|---|---|---|---|---|---|
| 1 | [0,4000] | [-4000,4000] | [1,50] | [1,10] | $[10^{-5},1]$ | $[10^{-10}, 10^{-5}]$ |
| 2 | [0,10000] | [-10000,10000] | [1,50] | [1,10] | $[10^{-5},1]$ | $[10^{-10}, 10^{-5}]$ |
| 3 | [4000,10000] | [-10000,10000] | [1,50] | [1,10] | $[10^{-5},1]$ | $[10^{-10}, 10^{-5}]$ |

the spectrum generator. A Universal Feynrules Output [56] file was manually written for MadGraph.[1] Micromegas 5.2.1 [58–61], which also uses the Sarah-generated files as an input, is used to compute the DM relic density $\Omega h^2$, spin-dependent and spin-independent DM cross sections for the proton and neutron, $\sigma^{SD}_{\tilde{\chi}^0_1,p}$, $\sigma^{SD}_{\tilde{\chi}^0_1,n}$, $\sigma^{SI}_{\tilde{\chi}^0_1,p}$, $\sigma^{SI}_{\tilde{\chi}^0_1,n}$ respectively, and the velocity-weighted DM-annihilation cross section $\langle\sigma v\rangle$, in addition to the active (co)-annihilation channels. The direct-detection limits are implemented using DDCalc 2.2.0 [62] using the most recent limits of Xenon [63, 64], PICO [65], LUX [66, 67], and PandaX [68, 69]. The spin-dependent and spin-independent cross sections are scaled with $(\Omega h^2)/0.12 \equiv \xi$ in order to account for any DM underabundance. Similarly, the velocity weighted cross section is scaled with $\xi^2$. The scaled values are used to interpret the direct-detection limits on $\Omega h^2$, and the indirect-detection limits on $\langle\sigma v\rangle$ from the Fermi-LAT gamma-ray limits of the Milky-way dwarf spheroidal galaxies [70] and the limits from IceCube and ANTARES on the neutrino limits from dark matter annihilation [29, 30], as these limits typically are produced with the assumption that only one single DM species exists. Furthermore, to obey limits from LHC searches, the $\sigma(pp \to \tilde{\chi}^0_2\tilde{\chi}^\pm_1)$, $\sigma(pp \to \tilde{\chi}^0_3\tilde{\chi}^\pm_1)$, and $\sigma(pp \to \tilde{\chi}^+_1\tilde{\chi}^-_1)$ are computed with MadGraph v3.1.1 [71] for all model points where the lightest neutralino mass $M_{\tilde{\chi}^0_1} \leq 350$ GeV. Cross sections including either $\tilde{\chi}^0_{4,5,6,7}$ or $\tilde{\chi}^\pm_2$ are ignored, since these will most likely not provide a significant signal if the $pp \to \tilde{\chi}^0_2\tilde{\chi}^\pm_1$, $pp \to \tilde{\chi}^0_2\tilde{\chi}^\pm_1$, or $pp \to \tilde{\chi}^+_1\tilde{\chi}^-_1$ processes did not do so already.[2] The LHC exclusion limits using the aforementioned production cross sections are determined with Smodels 2.0.0 [72–76].

The cuts in table 2 have been imposed on all generated model points as presented in section 4. The mass of the lightest chargino should be larger than 103.5 GeV, as charginos below this mass are excluded by LEP. The higgs sector has been expanded with two new higgs singlets, thus its mass computation obtains additional terms, introducing additional uncertainties. To be conservative, the SM-like higgs boson is required to have a mass between 120 and 130 GeV. The maximum variation of 5 GeV on the higgs mass is expected to have little impact on the DM candidates that we will focus on in what follows, the $\tilde{B}_{BL}$ and $\tilde{h}_{BL}$-like neutralinos. Furthermore, both the direct detection and LHC limits both need to be satisfied. Note that especially the cuts on the squark, slepton en gluino masses are more stringent than strictly

---

[1]This file can be obtained from Ref. [57].
[2]While the $\sigma(pp \to \tilde{\chi}^0_{4,5,6,7}\tilde{\chi}^\pm_1)$ may be higher, we ignore these scenarios, because searches will most likely not find these scenarios due to the complicated decay chain.

Table 2: Constraints that are imposed on all model points. The DM direct detection and the LHC production cross sections ($\sigma(pp \rightarrow \widetilde{\chi}_{2/3}^0 \widetilde{\chi}_1^\pm)$ and $\sigma(pp \rightarrow \widetilde{\chi}_1^+ \widetilde{\chi}_1^-)$) are implemented using DDCalc and Smodels respectively. The constraints on particle masses are implemented as a hard cut. Note that constraints on the sfermions and gluino masses are stricter than those provided by LHC searches, but this is done as so to guarantee that the model is not excluded by searches for coloured sparticles. Similarly, the $Z'$-boson mass is cut more stringently than required.

| Observable | Experiment |
|---|---|
| $\Omega h^2$ | Planck [77] |
| $\sigma_{SD,p}$ | PICO-60 [65] |
| $\sigma_{SD,n}$ | Xenon1T [63] |
| $\sigma_{SI}$ | Xenon1T [64] |
| $\langle \sigma v \rangle_{\chi\chi \rightarrow \gamma\gamma}$ | Fermi-LAT & HESS [70,78] |
| $\langle \sigma v \rangle_{\chi\chi \rightarrow \nu\nu}$ | IceCube & ANTARES [29,30] |
| $\sigma(pp \rightarrow \widetilde{\chi}_{2/3}^0 \widetilde{\chi}_1^\pm)$ | LHC [73] |
| $\sigma(pp \rightarrow \widetilde{\chi}_1^+ \widetilde{\chi}_1^-)$ | LHC [73] |
| 120 GeV $< m_h <$ 130 GeV | LHC [79] |
| $M_{\widetilde{\chi}_1^\pm} > 103.5$ GeV | LEP [80] |
| $m_{Z'} > 2.5$ TeV | LHC [81] |
| $m_{\widetilde{q}} > 2$ TeV | LHC [73] |
| $m_{\widetilde{l}} > 90$ GeV | LEP [82] |
| $m_{\widetilde{g}} > 2.5$ TeV | LHC [73] |

needed, but are chosen such that they easily evade the LHC limits and are factored out of the phenomenology. We deem these cuts reasonable for our purposes due to the limited impact of coloured sparticles on the DM sector.

We stress here that in the subsequent plots the number density of points is in no way indicative of any statistical significance, but simply a result of sampling; it is always possible to increase the sampling in the surrounding parameter space near low-density regions.

## 4 Results

### 4.1 Relic density

The relic density in the case of neutralino lightest stable particle (LSP) can be seen in figure 1. The vast majority of $\widetilde{W}$-like and $\widetilde{h}$-like LSPs are pure states. We consider a neutralino to be a pure state when the total fraction of any component exceeds 0.99. The pure states lie on a line with an $M_{\widetilde{\chi}_1^0}^2$ dependence due to the chargino-mediated annihilation channel. No $\widetilde{W}$ or $\widetilde{h}$ solutions are present below 103.5 GeV due to the LEP chargino limits,[3] thus the lightest chargino will have a mass similar to the lightest neutralino. The $\widetilde{B}_{BL}$ points, and to a lesser extent the $\widetilde{B}$ points, show a clear higgs funnel, as can be seen in figure 1 at $M_{\widetilde{\chi}_1^0} \approx 62.5$ GeV. Notably, there is no $Z$-boson funnel, as we tuned the particle-filter algorithm to search for $\widetilde{B}_{BL}$ and $\widetilde{h}_{BL}$ LSPs, which favour higgs and $Z'$ funnels. These funnels arise due to a higgsino component in both the $\widetilde{B}$ and $\widetilde{B}_{BL}$ LSPs. The $\widetilde{B}_{BL}$ neutralino couples to the SM-like higgs via

---

[3]Naturally wino and higgsino neutralinos with significant mixtures exist, but such neutralinos have not been optimised for explicitly as they are not the focus of this paper, and thus do not appear abundantly in the found solutions.

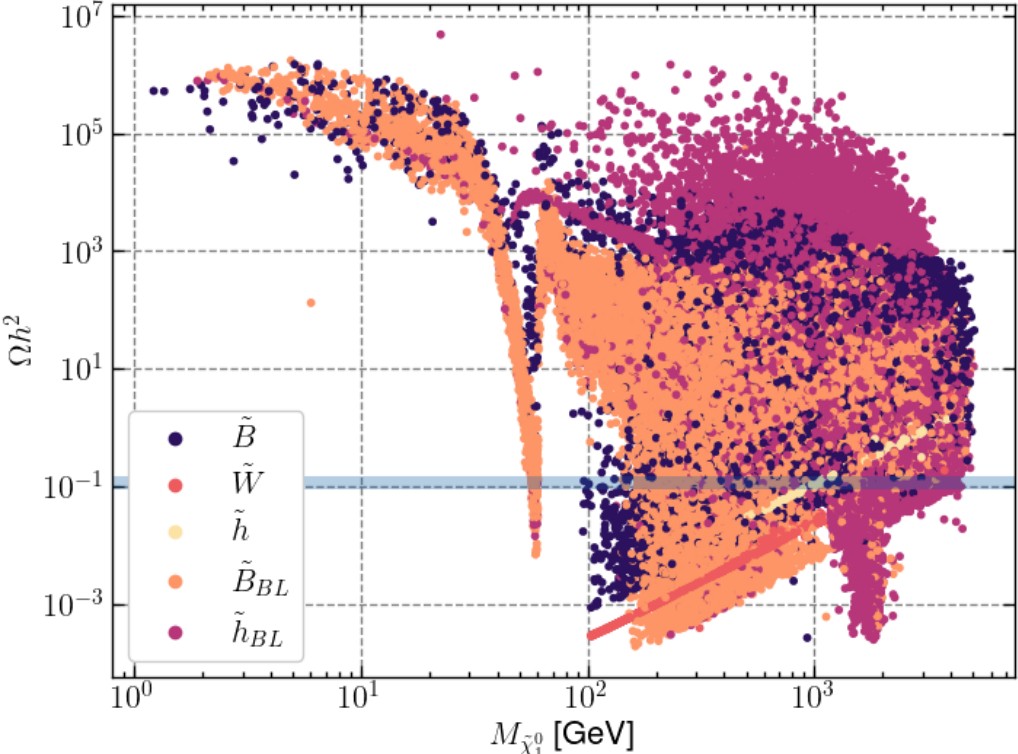

Figure 1: The relic density of the lightest neutralino as a function of its mass. The dominant component of the neutralino is colour coded as $\widetilde{B}$ (blue), $\widetilde{W}$ (red), $\widetilde{h}$ (yellow), $\widetilde{B}_{BL}$ (orange), and $\widetilde{h}_{BL}$ (purple). The observed relic density of the Planck collaboration [77] is shaded in blue including an uncertainty band of 0.03 to include computational/theoretical uncertainties. The model points shown here pass all constraints of table 2.

the $g_{\times}$ coupling constant. The $\widetilde{B}_{BL}$-$\widetilde{h}$ mixing term can clearly be seen in the neutralino mass matrix of Eq. (5). A funnel is also present for $\widetilde{B}$, $\widetilde{B}_{BL}$ and $\widetilde{h}_{BL}$ with the second lightest higgs $h_2$. However, the funnel cannot be seen in figure 1 since the mass of the second higgs is variable. Similarly a funnel for the third CP-even higgs $h_3$ and first CP-odd higgs $A_1^0$ exists, but these funnels are again not visible due to their variable masses. A funnel for $h_4$ and $A_2^0$ could not be found due to the high mass of both particles, since no solutions were found in which the mass of the first neutralino is approximately half that of $h_4$ or $A_2^0$. Moreover, many of the $\widetilde{B}$-like LSPs that result in a relic density that does not exceed the observed value are those where the first chargino and second neutralino have similar mass to the first neutralino. This enables a $\widetilde{\chi}_1^{\pm}$ and $\widetilde{\chi}_2^0$ (co)-annihilation mechanism in the early universe, thereby sufficiently lowering the relic density [83].

Figure 2 shows the $\widetilde{B}_{BL}$ (left) and $\widetilde{h}_{BL}$ (right) components for all LSPs that have passed the constraints of table 2, including a constraint on the relic density, $\Omega h^2 < 0.15$. Interestingly, of all $\widetilde{B}_{BL}$-like LSPs, none are pure $\widetilde{B}_{BL}$ states. This is expected from the neutralino mass matrix Eq. (5), which contains explicit mixing terms between the $\widetilde{B}$, $\widetilde{h}$, and $\widetilde{h}_{BL}$ fields. Similarly many $\widetilde{h}_{BL}$-like LSPs have a significant $\widetilde{B}_{BL}$ mixture. This high degree of mixture is caused by the relation between the mass of the $Z'$ boson and the mass of the LSP: the off-diagonal terms mixing the $\widetilde{B}_{BL}$ and $\widetilde{h}_{BL}$ fields in the neutralino mass matrix of Eq. (5) contain the terms $-g_{B-L}v_\eta$ and $g_{B-L}v_{\overline{\eta}}$, thus when the mass of a $\widetilde{h}_{BL}$ neutralino gets close to the that of the $Z'$ boson, these terms become relevant and mixing between $\widetilde{B}_{BL}$ and $\widetilde{h}_{BL}$ will occur. Of the found $\widetilde{h}_{BL}$-like LSPs the majority of the solutions with a relic density of 0.15 or lower have a funnel

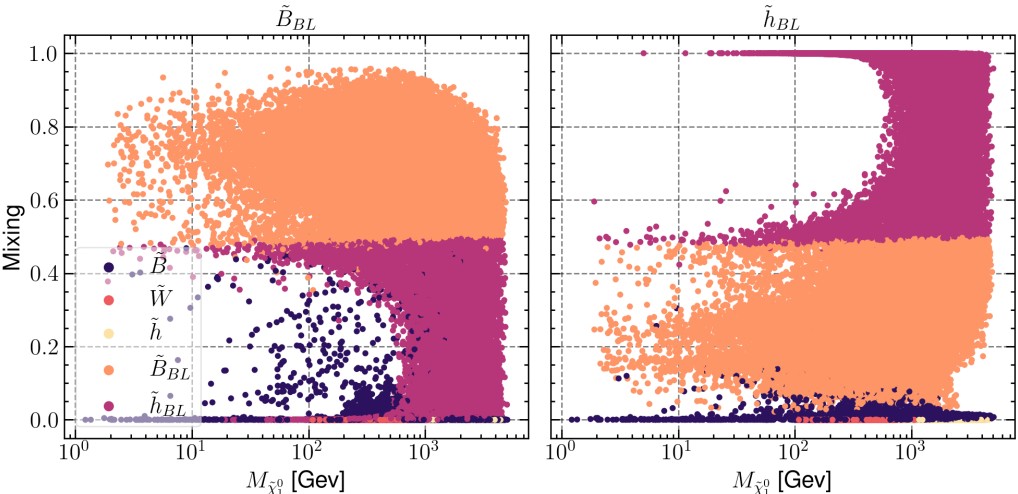

Figure 2: The size of the $\widetilde{B}_{BL}$ (left) and $\widetilde{h}_{BL}$ (right) components for the LSPs in all models that have passed the cuts of table 2, in addition to requiring $\Omega h^2 < 0.15$. Colour coding is the same as for figure 1.

of either the $Z'$, or a higgs kind. A direct consequence of this correlation and the cut $M_{Z'} > 2.5$ TeV is that the vast majority of the $\widetilde{h}_{BL}$ points with a good relic density are in the $M_{\widetilde{\chi}_1^0} > 1$ TeV region, since the $Z'$-boson mass is connected to the mass of the new higgses.

## 4.2   Direct detection limits

Figure 3 shows the spin-dependent and spin-independent cross sections, relevant for DM direct detection, of all non-excluded points with $\Omega h^2 < 0.15$.[4] The tentative LZ limits [84] are shown as a dotted line.[5] It can be seen that the SM-like Higgs funnel will be excluded, and the higgsino and $\widetilde{B}_{BL}$ region is probed. Additionally, the projected PICO limits for the spin-dependent cross section and the Xeno n-nT and Darwin limits for the spin-independent cross sections are shown. It can be seen that the projected limits on the spin-dependent cross section will not probe any of the found solutions. On the contrary, the projected spin-independent limits are expected to probe all of the found pure higgsino solutions. Similarly, the $\widetilde{B}_{BL}$-like LSP solutions will be within reach of Xenon-nT and DARWIN, but there remain solutions outside the reach of these experiments. It is therefore expected that, at least for the $\widetilde{h}_{BL}$ and some $\widetilde{B}_{BL}$-like LSPs, planned direct-detection experiments will not provide conclusive evidence on their potential existence.

For DM-nucleon scattering the value of the spin-dependent cross section is driven by a $t$-channel exchange of a $Z$ or $Z'$ boson, or a $t$-channel exchange of a squark. Similarly, the spin-independent cross section is driven by a mediating higgs, which can be any of the four CP-even higgs mass eigenstates, or again a mediating squark. Note, however, that the squark contribution to either the spin-dependent or spin-independent cross sections is often negligible due to their high masses of $\mathcal{O}$(2-10 TeV) for the found solutions, as the squark masses are cut at 2 TeV. The squark contributions become relevant for small $\widetilde{\chi}_1^0 \widetilde{\chi}_1^0 Z$ or $\widetilde{\chi}_1^0 \widetilde{\chi}_1^0 h$ couplings. However, for these scenarios, both the spin-dependent and spin-independent cross sections are much lower than the current or projected limits, so these models are not relevant for near-future DM phenomenology, and we will not discuss them further.

---

[4]Note that this includes a 0.03 uncertainty band is due to computational and theoretical, and not experimental, uncertainties.

[5]As of the writing of this paper the LZ limits have not yet been published in a peer-reviewed journal.

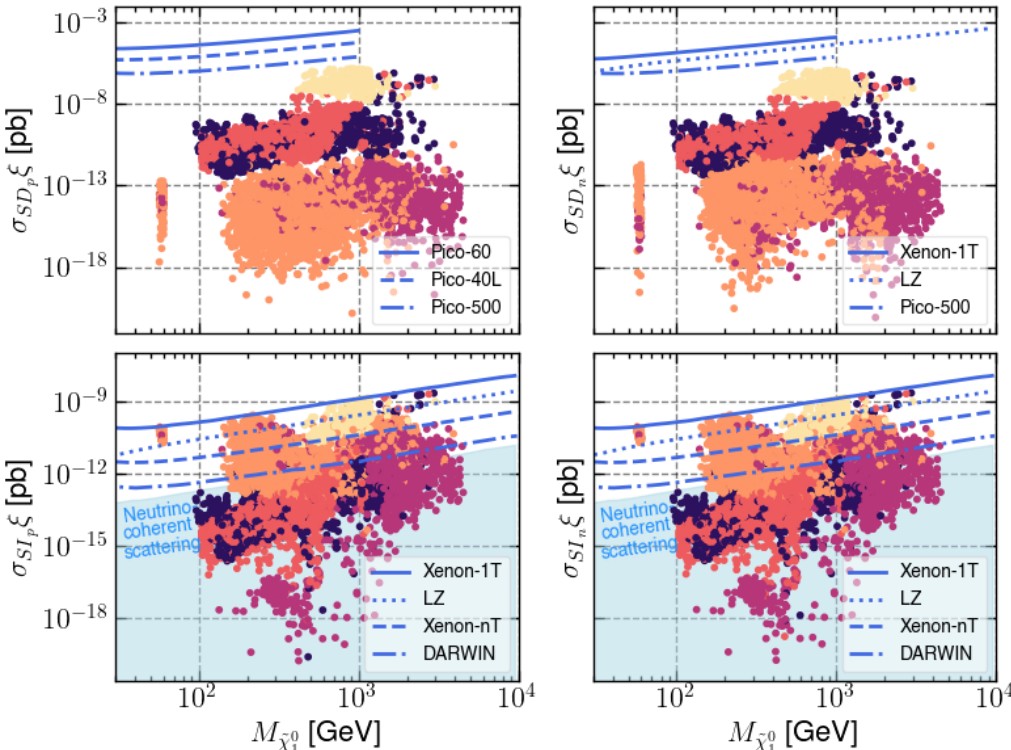

Figure 3: The spin-dependent (top) and spin-independent (bottom) DM cross sections for a target proton (left) or neutron (right) of all non-excluded neutralino LSP solutions as a function of the LSP mass. The cross section is given in units of picobarn (pb) and the mass is given in units of GeV. The tentative limits from LZ are shown in blue in addition to the projected limits from PICO, Xenon-nT, and DARWIN. Furthermore, the neutrino-coherent scattering floor is shown in shaded blue. The colour coding for the dominant contribution of the neutralino LSP is the same as in figure 1.

The higgsinos feature the largest spin-dependent cross section due to their coupling strength with the $Z$ boson. However, this coupling is suppressed for pure higgsino states since it is proportional to $N_{13}^* N_{13} - N_{14}^* N_{14}$, which for pure states becomes equal to zero. Notably, this suppression is not present for the $\tilde{\chi}_1^0 \tilde{\chi}_1^0 Z'$ coupling; while this coupling contains a similar term of the form $N_{16}^* N_{16} - N_{17}^* N_{17}$, the $\tilde{h}_\eta$ and $\tilde{h}_{\overline{\eta}}$ components are in most cases not the same, even for pure $\tilde{h}_{BL}$ states, due to the mixing terms between the $\tilde{B}_{BL}$ and $\tilde{h}_{BL}$ fields in the neutralino mass matrix. The bino and wino-like LSPs have a smaller coupling strength to the $Z$ boson compared to the higgsino LSPs, as these solutions by definition have a less higgsino sizable component compared to the higgsino neutralinos. The spin-dependent cross section for both the $\tilde{h}_{BL}$ and $\tilde{B}_{BL}$-like LSPs is generated via a mediating $Z$ and $Z'$. The $Z'$ contribution to the spin-dependent cross section for both the $\tilde{B}_{BL}$ and $\tilde{h}_{BL}$ has two dependencies: the amount of $\tilde{h}_{BL}$ present in the neutralino and the mass of the $Z'$. Given the heavy $Z'$-boson mass, the spin-dependent cross section for the $\tilde{B}_{BL}$ and $\tilde{h}_{BL}$-like LSPs is typically low.

Similarly to the spin-dependent cross section, the higgsino solutions also feature the highest values for the spin-independent cross section of all LSP types. Naturally, this results from their relatively large higgsino-bino-higgs and higgsino-wino-higgs coupling. However, in contrast to the spin-dependent cross section, both the $\tilde{B}_{BL}$ and $\tilde{h}_{BL}$ LSPs generally have higher cross sections than the wino LSPs. The cross section for models with a $\tilde{B}_{BL}$-like neutralinos is driven mainly by the SM-like higgs. The $\tilde{h}_{BL}$ neutralinos couple to the new $\eta$ and $\overline{\eta}$-like higgses as long as a sizeable amount of $\tilde{B}_{BL}$ is present, which is typically the case as seen from figure 2.

However, the size of the spin-independent cross section is suppressed by the masses of the new higgses, which can be heavy as they are driven by the $Z'$ mass, which in turn needs to be heavier than 2.5 TeV. The wino and bino solutions mostly have a low spin-independent cross section due to the correction factor on the direct detection cross sections of any relic density underabundance $\xi = \Omega h^2/0.12$.

## 4.3 LHC production cross sections

While the LHC phenomenology of the BLSSMIS is not focus of study in this paper, we here briefly comment on the typical production cross sections that our spectra predict. Figure 4 shows the size of the $pp \to \widetilde{\chi}_2^0 \widetilde{\chi}_1^{\pm}$ production cross section against the mass of the second neutralino, with the colour coding indicating the dominant contribution of the second neutralino. Unsurprisingly, two clear lines can be seen for a wino- and higgsino-like second neutralino, since the $pp \to \widetilde{\chi}_2^0 \widetilde{\chi}_1^{\pm}$ cross section is the largest for wino- or higgsino-like second neutralino and first chargino. However, when both the first and second neutralino have no sizable wino or higgsino component the cross section is not correlated to the mass of the second neutralino, which indeed can be observed in figure 4.

In the MSSM, the first or second neutralino is guaranteed to contain a sizeable wino or higgsino component or a mixture of both, which increases the chargino-neutralino production cross section. Moreover, in the MSSM the second neutralino cannot be a pure bino state, since the first neutralino then needs to be composed entirely of wino and higgsino components, which are excluded by either the relic density or dark matter direct detection experiments. This is no longer true in the BLSSMIS, as the $\widetilde{B}$, $\widetilde{B}_{BL}$, and $\widetilde{h}_{BL}$ components are able to compose the first four neutralino mass eigenstates, such that no large wino or higgsino component is

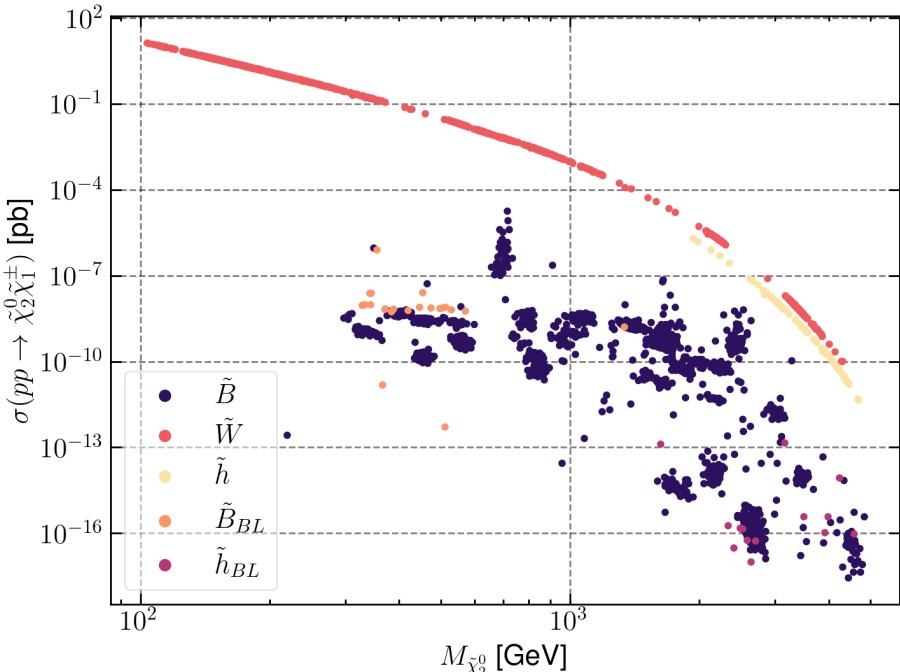

Figure 4: The $\sigma(pp \to \widetilde{\chi}_2^0 \widetilde{\chi}_1^{\pm})$ production cross section in pb against the mass of the second neutralino in GeV. The points are labelled according to the dominant contribution of the second neutralino. The shown points pass all constraints of table 2 and satisfy $\Omega h^2 < 0.15$.

present for these neutralinos. This results in the possibility of much lower production cross sections as compared to the MSSM. While the $\widetilde{h}_{BL}$-like first LSPs that pass the relic density cut are typically heavy, $\widetilde{B}_{BL}$-like LSPs can be low-mass. From figure 2 one can infer that low-mass $\widetilde{B}_{BL}$-like LSPs have sizable $\widetilde{B}$ and $\widetilde{h}_{BL}$ components. These scenarios typically translate to the lightest two neutralinos being composed of $\widetilde{B}$, $\widetilde{B}_{BL}$, and $\widetilde{h}_{BL}$, resulting in a potentially very low $pp \rightarrow \widetilde{\chi}_2^0 \widetilde{\chi}_1^\pm$ cross section. Thus the $\widetilde{B}$-like second neutralinos in figure 4 correspond mostly to $\widetilde{B}_{BL}$-like LSPs.

We have additionally computed both $\sigma(pp \rightarrow \widetilde{\chi}_3^0 \widetilde{\chi}_1^\pm)$ and $\sigma(pp \rightarrow \widetilde{\chi}_1^+ \widetilde{\chi}_1^-)$. Here $\sigma(pp \rightarrow \widetilde{\chi}_3^0 \widetilde{\chi}_1^\pm)$ is taken into account due to the aforementioned possibility of the first two neutralinos to not have a sizeable wino or higgsino component, and thus for the third neutralino to be the first wino/higgsino neutralino. Wino/higgsino neutralinos typically feature higher production cross sections than bino-like neutralinos. We disregard the neutralino-chargino production cross sections that include the fourth, fifth, sixth and seventh neutralino, as these most likely will not exclude a given model point if it has not been done so by the previously mentioned cross sections due to a lower cross section, an increased complexity of the neutralino decay chain, or both.[6] Since the focus of this paper lies on studying the phenomenology of our spectra at neutrino experiments, we refrain from commenting further on a possible LHC optimised search for these models.

## 4.4 Indirect-detection limits

From the previous subsections, we can conclude that most $\widetilde{h}_{BL}$-like DM solutions are not expected to be probed based on direct detection or collider experiments, and only future direct detection experiments may be sensitive to $\widetilde{B}_{BL}$ DM. This leaves indirect detection as a very interesting, and maybe only possible, short-term viable detection method. The scaled velocity-weighted cross section can be seen in figure 5. The Fermi-LAT limits from dwarf-spheroidal galaxies [70], HESS limits from the Galactic centre [78], and the IceCube and ANTARES [29, 30] limits on the resulting neutrino spectra are implemented, and for specific annihilation channels indicated by the solid blue lines in figure 5. In order to be conservative, we do not include the limits on the velocity-weighted annihilation cross section arising from antiproton limits due to large inherent hadronization and propagation uncertainties [19–27]. Furthermore, the projected KM3NeT limits for DMDM $\rightarrow \nu\nu$ is shown as a dotted line [85]. It can be seen that KM3NeT will be able to reach both the $\widetilde{\chi}_1^0 \widetilde{\chi}_1^0 \rightarrow \nu_l \nu_h$ and $\tilde{\nu}_h \tilde{\nu}_h$ regions.

The size of $\langle \sigma v \rangle \xi^2$ for $\widetilde{B}_{BL}$ and $\widetilde{h}_{BL}$-like DM is largely determined by the presence of either a $Z'$ or higgs funnel. If the corresponding DM masses are close to a $Z'$ or higgs resonance the velocity-weighted cross section increases rapidly. The $\widetilde{B}$, $\widetilde{W}$, and $\widetilde{h}$ neutralinos annihilate predominantly into SM particles; $W^+W^-$, leptons, quarks and the SM-like higgs. The branching ratios of neutralinos annihilating into the different final states depends on the exact composition of the neutralino, but only the branching ratios of annihilating neutralinos into neutrinos, either light or heavy, is of interest here. The annihilation spectra of the bino, wino and higgsino neutralinos will not be investigated further as these neutralinos annihilate mostly into SM particles, and the interest here lies with neutrino phenomenology that is not present in the MSSM. Both the $\widetilde{B}_{BL}$ and $\widetilde{h}_{BL}$-like DM can annihilate mostly (Br$[\widetilde{\chi}_1^0 \widetilde{\chi}_1^0 \rightarrow \nu\nu] > 0.5$) into neutrinos if either $\widetilde{\chi}_1^0 \widetilde{\chi}_1^0 \rightarrow \nu_h \nu_h$ or $\widetilde{\chi}_1^0 \widetilde{\chi}_1^0 \rightarrow \nu_h \nu_l$ is kinematically allowed. Both these processes are predominantly mediated via a $Z'$ boson,[7] and thus $\widetilde{B}_{BL}$ and $\widetilde{h}_{BL}$-like neutralinos

---

[6]In the case that the third neutralino is the first higgsino-like one, the fourth neutralino will very likely be also higgsino-like. Resultingly, the production of $pp \rightarrow \widetilde{\chi}_4^0 \widetilde{\chi}_1^\pm$ will be of a similar size to that of $pp \rightarrow \widetilde{\chi}_3^0 \widetilde{\chi}_1^\pm$. However, this will only increase the total production cross section by a factor of two, and is therefore only relevant for those spectra that feature production cross sections that are on the verge of detection. We deem these scenarios sufficiently unlikely that they can safely be neglected.

[7]Other mediating particles exist, but they are subleading in their contributions.

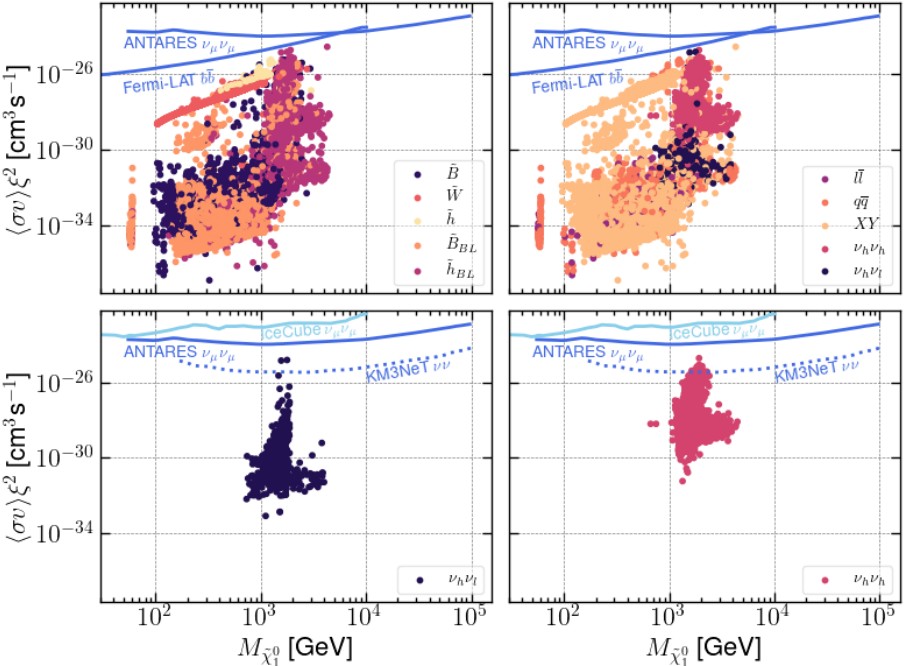

Figure 5: The DM velocity-weighted annihilation cross section (scaled with the square of their relic density divided by 0.12 and their branching ratio) in $\text{cm}^3\text{s}^{-1}$ as a function of the mass of the LSP (in GeV). The upper panels show the dominant component (upper-left) and dominant annihilation channel (upper-right). For the dominant annihilation channel, we use $q\bar{q}$ to indicate any combination of quarks produced in a DM annihilation process, $l\bar{l}$ any combination of leptons, $XY$ any combination of two bosons such that charge is conserved, $\nu_h \nu_h$ any combination of two heavy neutrinos, and $\nu_h \nu_l$ any combination of a heavy neutrino and light neutrino. The bottom panels isolate explicitly the LSPs that annihilate mostly into $\nu_h \nu_h$ (lower-right) and $\nu_h \nu_l$ (lower-left). The Fermi-LAT $b\bar{b}$ gamma-ray limit and ANTARES $\nu_\mu \nu_\mu$ neutrino limits are shown in the upper plots. The bottom plots show the IceCube and ANTARES $\nu_\mu \nu_\mu$ neutrino limits. Furthermore, the projected KM3NeT $\nu\nu$ limits are shown in the bottom plots. The cuts of table 2 have been applied to all shown points, and in addition we require $\Omega h^2 < 0.15$.

can annihilate mostly into neutrinos. Again, the specific branching ratios depend on the exact composition of the neutralino. Notably, for all neutralinos that annihilate mostly into $\nu_h \nu_l$ the process $\tilde{\chi}_1^0 \tilde{\chi}_1^0 \to \nu_h \nu_h$ is kinematically forbidden. Here $\nu_l$ and $\nu_h$ are the neutrino mass eigenstates as given in Eq. (7).

In total two relevant DM annihilation channels exist when regarding the neutrino spectra: $\tilde{\chi}_1^0 \tilde{\chi}_1^0 \to \nu_h \nu_h$, and $\tilde{\chi}_1^0 \tilde{\chi}_1^0 \to \nu_h \nu_l$. In general the process $\tilde{\chi}_1^0 \tilde{\chi}_1^0 \to \nu_h \nu_l$ process where the heavy neutrino $\nu_h$ subsequently decays is

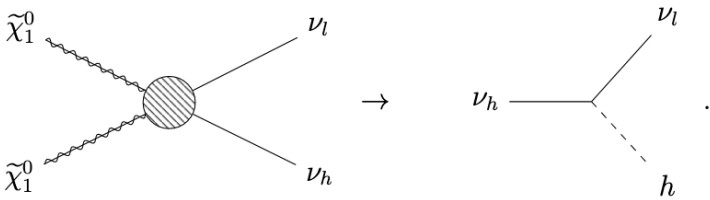

The momenta of the neutrinos resulting from neutralino annihilation can be computed using simple 2→2 kinematics. Moreover, the momenta of the neutrinos are completely fixed when assuming the velocity of the neutralinos to be negligible, which can safely be done seeing as the present-day DM particles are non-relativistic. For the process $\widetilde{\chi}_1^0 \widetilde{\chi}_1^0 \to \nu_h \nu_l$ the light neutrino $\nu_l$ then has an energy of

$$E_{\text{peak}} = \frac{(4M_{\widetilde{\chi}_1^0}^2 - M_{\nu_h}^2)}{4M_{\widetilde{\chi}_1^0}}, \tag{8}$$

where $M_{\widetilde{\chi}_1^0}$ is the mass of the neutralino and $M_{\nu_h}$ is the mass of the heavy neutrino. As the light neutrino has a single unique energy, rather than a distribution, a clearly defined peak in the neutrino spectrum must be present.

The heavy neutrino, via its decay products, of course also impacts the neutrino spectrum. In all cases studied the heavy neutrino has three main decay modes $\text{Br}[\nu_h \to \nu_l Z] \simeq \text{Br}[\nu_h \to \nu_l h] = \mathcal{O}(0.25)$ and $\text{Br}[\nu_h \to W^\pm l^\mp] = \mathcal{O}(0.5)$. The $Z$, $h$, $W^\pm$ and $l^\pm$ contribute to the neutrino spectrum via hadronisation and decays, and have a standard contribution to the neutrino spectrum. The contribution of SM particles to the total neutrino spectrum is best determined via Monte Carlo sampling. However, the contribution of the light neutrino can be computed analytically. In its rest frame, the heavy neutrino decays isotropically, thus the four-momenta of the two daughter particles are easily computed in general spherical coordinates in this frame. However, the heavy neutrino needs to be boosted from its rest frame to the rest frame of the annihilating neutralinos to determine the contribution of the light daughter neutrino to the total neutrino spectrum. When boosting in the $z$ direction[8] the energy of the daughter neutrino in the rest frame of the annihilating neutralinos is

$$E_{\text{plateau}} = \frac{M_{\nu_h}^2 - m_{h,Z}^2}{2M_{\nu_h}} (\cosh(\eta) + \sinh(\eta)\cos(\theta)), \tag{9}$$

$$\eta = \begin{cases} \cosh^{-1}\left(\frac{4M_{\widetilde{\chi}_1^0}^2 + M_{\nu_h}^2}{4M_{\widetilde{\chi}_1^0} M_{\nu_h}}\right) & \text{for} \quad \widetilde{\chi}_1^0 \widetilde{\chi}_1^0 \to \nu_h \nu_l, \\ \cosh^{-1}\left(\frac{M_{\widetilde{\chi}_1^0}}{M_{\nu_h}}\right) & \text{for} \quad \widetilde{\chi}_1^0 \widetilde{\chi}_1^0 \to \nu_h \nu_h. \end{cases}$$

Here $m_{h,Z}$ denotes the mass of the $Z$ or $h$ boson featuring in the decay $\nu_h \to \nu_l Z/h$. Note that the $\cos(\theta)$ term is from the momentum of the light neutrino arising from the decay of the heavy neutrino in the $z$ direction in the rest frame of the heavy neutrino. Thus the energy of the light neutrino depends on the angle $\theta$, which implies that the contribution of this neutrino to the total neutrino spectrum has an energy range that depends on $\theta$. It should be noted that, since the general four momenta of the neutrinos are written in terms of spherical coordinates, the angle $\theta$ needs to be sampled according to $\cos^{-1}(1-2u)$ with $u \in [0,1]$ to get a uniform distribution of points on a sphere,[9] as is required by the isotropy of the decay of the heavy neutrino. This causes the energy spectrum of the light neutrinos coming from heavy neutrino decay to lie on a flat line, i.e. it will form a plateau. The edges of the plateau correspond to $\cos(\theta) = 0$ for the high end and $\cos(\theta) = -1$ for the low end as can be seen from Eq. (9). The number of neutrinos in the peak and plateau regions is given by

$$N_{\nu,\text{peak}} = \text{BR}[\widetilde{\chi}_1^0 \widetilde{\chi}_1^0 \to A\nu_l]/N_f, \tag{10}$$

$$N_{\nu,\text{plateau}} = \text{BR}[\widetilde{\chi}_1^0 \widetilde{\chi}_1^0 \to \nu_h A] \cdot \text{BR}[\nu_h \to \nu_l B]/N_f. \tag{11}$$

---

[8]The boost direction can of course freely be chosen. The $z$ direction is chosen here, since it yields the simplest expression for the neutrino energy as by convention the $z$ component in spherical coordinates only has a $\cos(\theta)$ term.

[9]See http://corysimon.github.io/articles/uniformdistn-on-sphere/ for an in-depth explanation.

Here $N_{\nu,\text{peak}}$ and $N_{\nu,\text{plateau}}$ are the number of neutrinos in the peak and plateau regions respectively. Furthermore, $A$ and $B$ are used to indicate any particle flavour (including heavy/light neutrinos) and $N_f$ the number of light neutrino flavours. Note that if $A = \nu_h$ the number of neutrinos in the plateau is twice as large compared to the case where $A \neq \nu_h$.

In figure 6 the neutrino spectra of two example spectra (whose LSP and heavy neutrino masses, plateau and peak energies are shown in table 3) are shown that have as the dominant annihilation channel $\widetilde{\chi}_1^0 \widetilde{\chi}_1^0 \to \nu_h \nu_h$ on the left and $\widetilde{\chi}_1^0 \widetilde{\chi}_1^0 \to \nu_h \nu_l$ on the right. The predicted peak and plateau regions are indicated by the shaded regions. Both features can clearly be seen in the computed spectra. The peak region can be seen at $E \approx 950$ GeV for the $\nu_h \nu_h$ channel and at $E \approx 300$ GeV for $\nu_h \nu_l$. The plateau is visible between $300 \lesssim E \lesssim 850$ GeV for $\nu_h \nu_h$ and $500 \lesssim E \lesssim 800$ GeV for $\nu_l \nu_l$. Note that the peak for the $\nu_h \nu_h$ dominant channel in figure 6 is due to the presence of a sub-dominant annihilation channel $\widetilde{\chi}_1^0 \widetilde{\chi}_1^0 \to \nu_h \nu_l$ which has a branching ratio of approximately 1%. The computed values of the peak and plateau regions are shown in table 3. When compared to the spectra of figure 6 these correspond precisely to the `MadGraph` data,[10] up to binning effects.

Table 3: The numerical values of the energy of the peak ($E_{\text{peak}}$), the low-energy ($E_{\text{plateau low}}$) and high-energy ($E_{\text{plateau high}}$) boundary of the plateau of the neutrino spectrum for the two show-case files in GeV.

| Main channel | $m_{\widetilde{\chi}_1^0}$ (GeV) | $m_{\nu_h}$ (GeV) | $E_{\text{plateau low}}$ (GeV) | $E_{\text{plateau high}}$ (GeV) | $E_{\text{peak}}$ (GeV) |
|---|---|---|---|---|---|
| $\widetilde{\chi}_1^0 \widetilde{\chi}_1^0 \to \nu_h \nu_h$ | 1208 | 1083 | 332 | 860 | 965 |
| $\widetilde{\chi}_1^0 \widetilde{\chi}_1^0 \to \nu_h \nu_l$ | 830 | 1311 | 513 | 822 | 312 |

Additionally, the height of the peak of the right (left) plot in figure 6 can be seen to be $\sim 0.3$ ($\sim 0.0026$), which is as predicted by Eq. (10) since $\text{BR}[\widetilde{\chi}_1^0 \widetilde{\chi}_1^0 \to \nu_h \nu_l]$ is $\sim 0.96$ ($\sim 0.01$) and $\text{BR}[\nu_h \to \nu_l B] \approx 0.5$. Similarly, the total number of neutrinos in the plateau of the right (left) plot is $\sim 0.16$ ($\sim 0.36$), which is again as expected based on Eq. (11). From these two example spectra, it can be verified that the shape and size of the peak and plateau regions can accurately be predicted given the model parameters. Moreover, the shape of the peak and plateau regions is uniquely specified. Thus any possible future measurement will directly provide insight into the neutralino mass, the heavy neutrino mass, the branching fractions of a heavy neutrino, and the branching fraction of two annihilating neutralinos into $\nu_h \nu_h$ and $\nu_h \nu_l$. We however, refrain from giving exclusion lines on this spectrum. While the peak of the spectrum can be directly tied to the monochromatic lines of $\nu_\mu \nu_\mu$ exclusion limits, namely the exclusion limit for $\tilde{\chi}_1^0 \tilde{\chi}_1^0 \to \nu_l \nu_h$ is half that of $DMDM \to \nu_{e,\mu,\tau} \nu_{e,\mu,\tau}$ at the worst, the impact of the plateau and remaining features of the spectrum in conjunction with each other requires experiment-specific analysis, especially the detector response. Especially the impact of the plateau is important, as detecting a peak does not, at least in this model, completely fix the dark matter mass. Thus a possible neutrino signal from dark matter may differ from usual assumptions.

## 5  Conclusion

Neutrino detection experiments have gained a significant increase in sensitivity over the past few years. It becomes therefore interesting to study their ability to try and detect neutrinos

---

[10]We use `MadGraph` to simulate $\chi \chi$ annihilating to all particles at leading order with "beam energy" equaling to $m_\chi$, and use `PYTHIA` [86] to simulate the hadronization and decay. Finally, all events with neutrino pairs in final states are selected.

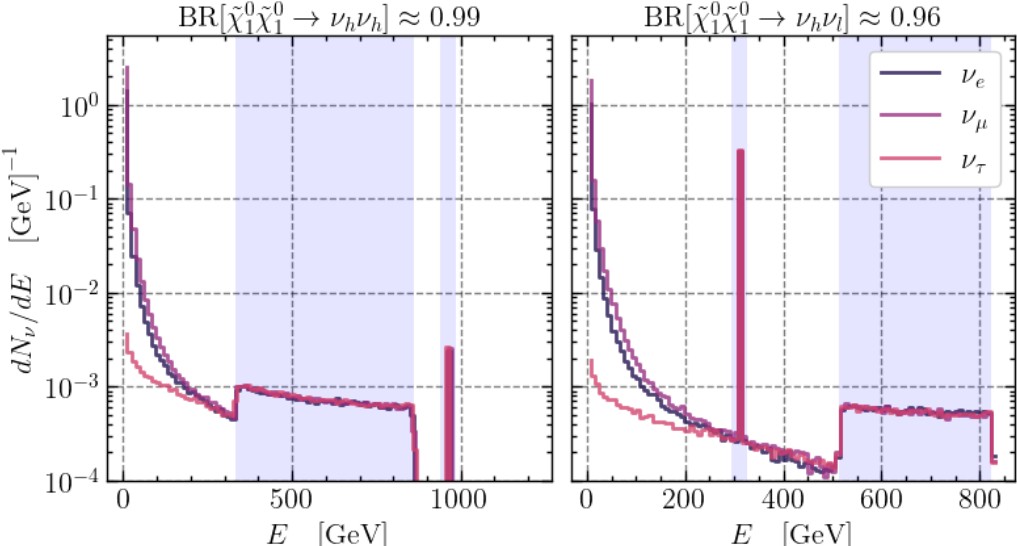

Figure 6: The neutrino spectrum of two main different annihilation channels: $\widetilde{\chi}_1^0 \widetilde{\chi}_1^0 \to \nu_h \nu_h$ (left) and $\widetilde{\chi}_1^0 \widetilde{\chi}_1^0 \to \nu_h \nu_l$ (right). The three different neutrinos $\nu_e$, $\nu_\mu$, $\nu_\tau$ have been colour coded. All spectra have been computed by `MadGraph` using 100k events. The predicted peak and plateau regions have been shaded. The width of the shade for the peak region has been widened for visibility.

originating from DM annihilations in the present-day universe. In this paper, we have examined one model that allows for such annihilations, the BLSSMIS. A particular difficulty so far in probing the phenomenology that follows from this model is that it features DM particles that couple only very moderately to SM particles. Such DM particles typically have a dominant $\widetilde{B}_{BL}$ or $\widetilde{h}_{BL}$ component. The resulting DM direct detection cross sections for such particles are well below the neutrino-coherent scattering limit, and the accompanying spectra typically involve neutralino-chargino production cross sections at the LHC that are much lower than those found in the MSSM. However, we find that precisely for these spectra, indirect detection by means of neutrino detection experiments can be used to probe them. The scattering of high-mass $\widetilde{B}_{BL}$ or $\widetilde{h}_{BL}$-like DM particles in our present-day universe can create $\nu_h \nu_h$ ($\nu_h \to \nu_l + X$, where $X$ is any other SM particle except one of the lightest neutrinos, $\nu_l$) and $\nu_h \nu_l$ final states. We have shown that a typical feature of such annihilation channels is that they predict a plateau region and a single peak in the neutrino-energy spectrum, which are distinct features that can be measured using neutrino telescope experiments. The extent of the peak region and the location of the energy peak are both completely specified by the mass of the heavy neutrino and that of the DM particle. This shows that measurements of the energy spectrum of cosmic neutrinos can provide a clear and unique way of discovering DM.

# Acknowledgements

MvB acknowledges support from a Royal Society Research Professorship (RP\R1\180112), and by the Science and Technology Facilities Council (ST/T000864/1). R. RdA acknowledges the Ministerio de Ciencia e Innovación (PID2020-113644GB-I00).

# A  Details of the BLSSMIS Model

In this appendix, we offer more details about the charges of the chiral superfields in the B-L-SSM-IS model and terms in the superpotential (1). Beyond the MSSM, 5 chiral superfields are introduced: $\hat{\nu}$, $\hat{\eta}$, $\hat{\bar{\eta}}$, $\hat{s}_1$, and $\hat{s}_2$. There are two ways to assign their gauge-group charges. We show the detailed charge numbers in table 4 for assignment (I) and (II). The superfield charges in the MSSM sectors are the same for both assignments, while the extended sectors in B-L-SSM-IS models are assigned different charges, with the exception of $\hat{\nu}$, which must have the charge asignment of a right-handed neutrino.

For both assignments, we integrate $\hat{s}_1$ out so all terms with $\hat{s}_1$ vanish, thereby only contributing to anomaly cancellations. For undetectable terms, setting their couplings to 0 is safe for phenomenology research, as we always need to consider the simplicity of our models such that the remaining couplings and sectors are testable.

Additionally, for assignment (I), i.e. the one used for this study, $\mu_S \hat{s}_2 \hat{s}_2$ explicitly breaks B-L symmetry, while an allowed term $Y_S \hat{\eta} \hat{s}_2 \hat{s}_2$ is neglected in Eq. 1. On the other hand, for assignment (II) $\mu_S \hat{s}_2 \hat{s}_2$ and $Y_S \hat{s}_2 \hat{s}_2 \hat{s}_2$ are allowed, while $Y_S \hat{\eta} \hat{s}_2 \hat{s}_2$ is forbidden. Similarly to the treatment for $\hat{s}_1$, we in general want to avoid undetectable sectors and hence set $Y_S = 0$ in both assignments. Furthermore, although the second assignment agrees with B-L symmetry, we have to manually set the coupling $\mu_S$ to be very small in order for the inverse see-saw mechanism to work. In contrast, the term $\mu_S \hat{s}_2 \hat{s}_2$ that explicitly breaks B-L symmetry in assignment (I) is assumed to be generated from higher order effects automatically. Therefore, $\mu_S$ should be small enough automatically. This is the main reason why this study and most of the other published studies focus on assignment (I), instead of (II).

Finally, to achieve the model files for Micromegas and `MadGraph`, we use the source code in SARAH database.[11] The assignment (I) is the original version in Sarah database, while we edit a modified version to realize assignment (II). The charges and superpotential are set in file `B-L-SSM-IS.m`, we modify it according to the values in Table 4 and regenerate SPheno and UFO files.

Table 4: Chiral Superfields of B-L-SSM-IS.

| Superfield | $U(1)_Y$ | $SU(2)_L$ | $SU(3)_C$ | $U(1)_{B-L}$ (I) | $U(1)_{B-L}$ (II) |
|---|---|---|---|---|---|
| MSSM | | | | | |
| $\hat{H}_u$ | $\frac{1}{2}$ | **2** | **1** | $0$ | $0$ |
| $\hat{H}_d$ | $-\frac{1}{2}$ | **2** | **1** | $0$ | $0$ |
| $\hat{q}$ | $\frac{1}{6}$ | **2** | **3** | $\frac{1}{6}$ | $\frac{1}{6}$ |
| $\hat{u}$ | $-\frac{2}{3}$ | **1** | **3̄** | $-\frac{1}{6}$ | $-\frac{1}{6}$ |
| $\hat{d}$ | $\frac{1}{3}$ | **1** | **3̄** | $-\frac{1}{6}$ | $-\frac{1}{6}$ |
| $\hat{l}$ | $-\frac{1}{2}$ | **2** | **1** | $-\frac{1}{2}$ | $-\frac{1}{2}$ |
| $\hat{e}$ | $1$ | **1** | **1** | $\frac{1}{2}$ | $\frac{1}{2}$ |
| Extension | | | | | |
| $\hat{\nu}$ | $0$ | **1** | **1** | $\frac{1}{2}$ | $\frac{1}{2}$ |
| $\hat{\eta}$ | $0$ | **1** | **1** | $-1$ | $-\frac{1}{2}$ |
| $\hat{\bar{\eta}}$ | $0$ | **1** | **1** | $1$ | $\frac{1}{2}$ |
| $\hat{s}_1$ | $0$ | **1** | **1** | $-\frac{1}{2}$ | $0$ |
| $\hat{s}_2$ | $0$ | **1** | **1** | $\frac{1}{2}$ | $0$ |

---

[11] https://sarah.hepforge.org/trac/attachment/wiki/B-L-SSM-IS/B-L-SSM-IS.tar.gz.

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
