# Peer review of "Non-Standard Neutrino Spectra From Annihilating Neutralino Dark Matter"

_SciPost Physics Core, doi:SciPost Phys. Core 6, 006 (2023)_

## Round 2 · Referee Report · Anonymous (Referee 6) · 2022-9-15

Report

My report is provided in the attached pdf, and outlines the strenghts,
weaknesses and requested changes.

Attachment

  • validity: ok
  • significance: ok
  • originality: ok
  • clarity: low
  • formatting: acceptable
  • grammar: good

Author:  Jochem Kip  on 2022-10-11  [id 2911]

(in reply to Report 1 on 2022-09-15)

Best,

We thank you for your report, we would of course like to comment on some of your points and address some concerns you have.

Regarding the weaknesses:
-We have added lines showing the exclusion limit for dark matter annihilating directly into neutrinos, and added a paragraph discussing its impact. We, however, deem an in-depth analysis of the plateau region unviable, as it would require an experiment-specific detector response to see what its possible impact and detectability would be.

-The included error of 0.03 on Ωh2 is done so in order to take theoretical uncertainties into account. We have clarified this in the text. The experimental uncertainty is indeed much smaller than 0.03. We use an uncertainty of 0.03 on Ωh2 in order to account for slightly different cosmologies. Especially since a tighter bound on Ωh2 will not change our conclusions in any meaningful way. This uncertainty is fairly commonplace in similar analyses.

-We have hopefully clarified the scanning objective in the text.

-We have included relevant citations in the introduction.

Regarding the requested changes:
-We have expanded upon the neutrino telescope analysis, hopefully to a satisfactory degree.

-The t-channel exchange of sleptons does indeed exist, but this channel is subleading with respect to the Z’ s-channel. We have updated the text to mention this. We do note that the specific channel is irrelevant for the kinematics of the neutrino spectra, this has also been updated in the text.

-More information on the sampling procedure has been added.

-In figure 5 the fraction of the dominant annihilation channel is shown. The description of the text has been updated to reflect this. Our thanks for notifying us to this oversight.

-If one were only to split MBL and not M1, M2, and M3 at the GUT scale, the phenomenology at the LHC and DD experiments would be impacted, as the fraction of bino, wino, and higgsino is of great importance. Thus in order to take the scenarios into account we have chosen to use four gaugino parameters as input. One could additionally argue that splitting MBL from M1/2 at the GUT scale, but not M1, M2, and M3, is artificial.
Regarding the sampling of μS, there are two possible scenarios: either it influences the spectrum and one should scan over it, or it does not, and then it does not impact the scanning procedure since all parameters are sampled independently in a Gaussian particle filter. Thus we have opted to use μS as a free parameter.

-The recent LZ exclusion bounds have been included, however, not as a strict exclusion limit as the results are not yet published in a peer-reviewed journal.

-The corrections on μS are small because it is a symmetry-breaking parameter as it breaks B-L symmetry. Therefore it can only receive high-scale radiative corrections which must be small. The text has been updated to hopefully clarify this, including a relevant citation.

-We have added a paragraph briefly discussing previous work on the sneutrino in this model and why we do not investigate this further. We thank the referee for pointing out that this paragraph was absent.

Kind regards,
The authors

---

## Round 2 · Referee Report · Anonymous (Referee 5) · 2022-9-16

Strengths

1. The model building looks accurate and complete.
2. The analysis is correct.

Weaknesses

1. The chosen model/framework is not particularly compelling nor timely.
2. The phenomenological analysis lacks depth and does not consider all the aspects.

Report

The paper deals with the phenomenology of a supersymmetric construction that provides a Dark Matter candidate. In particular, the paper focuses on indirect detection searches. The presentation is twisted to be interesting for neutrino telescopes, in that it produces an unconventional spectrum in those, while passing other direct and indirect constraints, but (as discussed below) in the end the analysis of this part is not particularly deep.

I don't feel fully competent in judging the supersymmetric model building aspects, so I hope that another referee will cover these sides. To the non-expert eye, the model looks reasonable, even if rather elaborate and not particularly compelling. It is also somewhat out of sync with respect to the current trends in the field, in which Susy constructions (extensions of the MSSM) of this sort have almost disappeared.

The phenomenological analysis, in particular concerning indirect detection (which is the main focus of the paper), lacks depth and is not particularly compelling either. The authors consider the constraints from Fermi dwarfs, but they have neglected the more stringent ones coming from Imaging Atmospheric Cerenkov Telescopes such as HESS (see e.g. arXiv:2108.13646). In neutrinos, which again are the central focus of the analysis, the authors have considered ICECUBE but neglected Antares (see e.g. arXiv:1612.04595). they have also completely neglected charged cosmic rays, in particular antiprotons, which are instead known to produce relevant constraints in this mass range (see. e.g. arXiv:1610.03071, arXiv:1712.00002 and subsequent work).

In summary, in my opinion the paper does not qualify for a publication on SciPost. The analysis is most probably correct, and clearly has required significant work, but unfortunately the considered model does not look compelling, not timely and the phenomenology lacks exhaustiveness. It seems to me a somewhat useful addition to the literature, but not a particularly relevant one.

  • validity: good
  • significance: good
  • originality: ok
  • clarity: high
  • formatting: excellent
  • grammar: excellent

Author:  Jochem Kip  on 2022-10-11  [id 2910]

(in reply to Report 2 on 2022-09-16)

Best,

We thank you for your report, we would of course like to comment on some of your points and address some concerns you have

-We have expanded somewhat upon the neutrino spectrum analysis, especially regarding the peak of the spectra. However, we deem an in-depth study of any possible exclusion limit of these specific spectra to be outside of the scope of this work, as it would require detector-specific response studies.

-Regarding the HESS limits, as seen in Figure 1 of arXiv:2108.13646, the limits as provided from HESS are orders of magnitude weaker than those given by Fermi-LAT for XX-> bb, and only at ~10^4 do they become more stringent than Fermi-LAT for XX-> tau tau. However, there are no points in this region, thus we have not implemented these limits.

-The neutrino limits are taken from both a combined search of IceCUBE and ANTARES and both experiments separately. This fact is now accurately mentioned in the text. We thank the referee for notifying us to this oversight.

-We have indeed not implemented the limits from charged cosmic rays as reported by various papers, as recent studies have shown that large uncertainties exist in cosmic ray propagation models, which have not previously been taken into account. These limits are also not particularly stringent, and will not change the main conclusions of the paper.

Kind regards,
The authors

---

## Round 2 · Referee Report · Anonymous (Referee 4) · 2022-9-17

Report

In this work the authors consider a specific extension of the MSSM that
incorporates the inverse seesaw mechanism to generate neutrino masses.
A possible candidate for the cosmological dark matter (DM) is then the
lightest neutralino -- which in this model features a 7x7 mass matrix
rather than the usual 4x4 matrix encountered in the MSSM.
The focus of this article is a specific part of the parameter space of
this model, where DM dominantly annihilates into neutrinos and other
searches would have very little constraining power. As the authors point
out, the neutrino spectrum from DM annihilation can be rather peculiar in
these cases, which may allow to discriminate such scenarios from neutrino
signals in more conventional DM scenarios.

I mainly have two concerns:

1) Scatter plots, and in particular the density of points in these plots,
lack any well-defined statistical significance. The authors should thus
strongly consider to indicate contours for the parameter space they consider,
rather than showing points distributed with non-uniform density (unless
they actually combine all constraints in a statistical meaningful way).
Alternatively, the authors should explicitly stress that the density of
points cannot directly be connected to any form of statistical significance.
(In fact, with one of the authors also being a coauthor of 2012.09874,
I wonder why this is is not even mentioned as a complication.) There is in
fact also the issue that sometimes it is impossible to see how many points
of different style are on top of each other (e.g. in Fig.), so further panels
might be warranted. Finally, any mentioning of "many", "(vast) majority of",
"typical" etc. must be removed when discussing the models represented
in these plots.

2) Fig. 6 is a nice illustration of spectra that are indeed rather specific.
In order to back conclusions like "This shows that indirect-detection
experiments may be used as a powerful probe for these models" there are two
more things that must be demonstrated:
a) do the spectra remain as strikingly different from standard spectra when
the rather poor (compared to, e.g., gamma rays) energy resolution of
neutrino telescopes is taken into account?
b) What about total rates -- are they competitive for neutrino signals from
the Galactic center or halo ? (neutrino signals from the center or the Earth
or Sun would not be competitive, I'd presume, because the scattering rates
are so low. The authors should in any case comment on this!)

More minor comments:

3) Also the MSSM does not necessarily produce "vanilla" neutrino spectra from
neutralino annihilation, as (at least indirectly) claimed; see, e.g., 1705.03466
for examples of very characteristic deviations from this.

4) Table 2 lists the "hard cuts" employed in the scan. Do these limits all
correspond to published constraints at the same, or similar, confidence levels?
Also, why does the relic density constraint not appear in this list -- and why is
it set to oh2 < 0.15 (if I understand correctly)), which would correspond to a much
weaker limit (in terms of C.L.) than the other limits.

5) I was at first very confused about the placement of Fig 1, so please consider
moving it to after the first mentioning in the text.

  • validity: good
  • significance: ok
  • originality: ok
  • clarity: high
  • formatting: good
  • grammar: good

Author:  Jochem Kip  on 2022-10-11  [id 2909]

(in reply to Report 3 on 2022-09-17)

Best,

We thank you for your report and would like to reply/address some of the points you have raised.

Regarding your major concerns:
-You are of course completely correct that the density of points does not correspond to any statistical significance, especially because the search is performed in order to find new phenomenology. We have updated the text to clearly reflect this. Furthermore, any mention of “majority of”, “typical”, etc. have been removed.

-Whether the spectra remains strikingly different from standard spectra is a particularly difficult question to answer. The peak can of course be directly tied to monochromatic neutrino spectra, which we have updated and mentioned in the text. It would be safe to assume that the plateau will be different from the usual annihilation spectra. The specifics would require experiment-specific detector response studies, which we deem to be outside the scope of this paper. Of course, one would indeed expect that the clearly defined shapes of the neutrino spectra as shown in Figure 6 will not translate as clearly to neutrino telescopes. However, the purpose of this paper is also to make experimentalists aware that these non-standard spectra exist.
Regarding the rates of the neutrino flux, we have included the projected KM3NeT limits, which show that the new spectra will be in reach of KM3NeT.

Regarding your minor comments:
-The text has been updated to mention that the neutrino annihilation spectra resulting from the MSSM can deviate from “vanilla” spectra.

-The experimental limits all correspond to a 90-95% C.L., with some exceptions, e.g. the squarks and gluinos are required to have a high mass, which is oftentimes more stringent than strictly needed. This is due to the objective of finding new phenomenology, not necessarily exploring the complete parameter space. The uncertainty on Ωh2 of 0.03 is due to theoretical uncertainties. It is difficult to explicitly set a C.L. to this number, however, as many of our conclusions will not change with a more stringent limit on Ωh2, we have opted for this uncertainty in order to be conservative, similar to the uncertainty on the mass of the SM-like Higgs boson. Furthermore, the limit of Ωh2 has been added to Table 2.

-Figure 1 has been moved.

Kind regards,
The authors

---

## Round 3 · Referee Report · Anonymous (Referee 2) · 2022-11-7

Report
I thank the authors for their response and for the modifications that they have made to the paper.
Unfortunately, I hold to my assessment that the paper does not qualify for publication on SciPost.
In the following I elaborate on the comments by the authors in response to my remarks.
The authors have written:
-We have expanded somewhat upon the neutrino spectrum analysis, especially regarding the peak of the spectra. However, we deem an in-depth study of any possible exclusion limit of these specific spectra to be outside of the scope of this work, as it would require detector-specific response studies.
Barring mistakes on my side, the modifications to the text in v3, with respect to v2, are really minimal: two sentences in the Introduction (of which the first doesn't read well: missing a verb?); one sentence at the end of Sec. 2; two sentences in Sec. 4.2; one sentence and the end of Sec. 4.4. The discussion of the spectrum does not look deeper than before. The figure just above eq.(8) is actually less clear than before.
The authors have written:
-Regarding the HESS limits, as seen in Figure 1 of arXiv:2108.13646, the limits as provided from HESS are orders of magnitude weaker than those given by Fermi-LAT for XX-> bb, and only at ~10^4 do they become more stringent than Fermi-LAT for XX-> tau tau. However, there are no points in this region, thus we have not implemented these limits.
There has been a misunderstanding here and I apologize for the confusion I induced. I was referring to the constraints by HESS (and other IACTs) coming from observations of the Galactic Center, not from dwarfs. The constraints from the GC area may be relevant and apply to masses within the range considered in the paper. See for instance, for the historical evolution and the current status, and restricting only to HESS: arXiv:1103.3266, 1509.04123, 1607.08142, 2207.10471.
By mistake I pointed out the reference 2108.13646, which reports the constraints from HESS from dwarfs. These indeed are not competitive, in the current context. I apologize for that.
The authors have written:
-We have indeed not implemented the limits from charged cosmic rays as reported by various papers, as recent studies have shown that large uncertainties exist in cosmic ray propagation models, which have not previously been taken into account. These limits are also not particularly stringent, and will not change the main conclusions of the paper.
This response is not particularly satisfying. The authors should substantiate their statements more soundly.
First, to my knowledge uncertainties have been taken into account since a long time, and they have gone down significantly recently.
Second, what is the basis by which the authors claim that the limits are not stringent? They should have cited a reference or an estimate. For instance, the constraints based on AMS antiprotons (see fig 13, upper panel, of 1712.00002, the reference provided in the first report) may have an impact, if one compares with Fig. 5 of the present paper.
Unfortunately, I hold to my assessment that the paper does not qualify for publication on SciPost.
In the following I elaborate on the comments by the authors in response to my remarks.
The authors have written:
-We have expanded somewhat upon the neutrino spectrum analysis, especially regarding the peak of the spectra. However, we deem an in-depth study of any possible exclusion limit of these specific spectra to be outside of the scope of this work, as it would require detector-specific response studies.
Barring mistakes on my side, the modifications to the text in v3, with respect to v2, are really minimal: two sentences in the Introduction (of which the first doesn't read well: missing a verb?); one sentence at the end of Sec. 2; two sentences in Sec. 4.2; one sentence and the end of Sec. 4.4. The discussion of the spectrum does not look deeper than before. The figure just above eq.(8) is actually less clear than before.
The authors have written:
-Regarding the HESS limits, as seen in Figure 1 of arXiv:2108.13646, the limits as provided from HESS are orders of magnitude weaker than those given by Fermi-LAT for XX-> bb, and only at ~10^4 do they become more stringent than Fermi-LAT for XX-> tau tau. However, there are no points in this region, thus we have not implemented these limits.
There has been a misunderstanding here and I apologize for the confusion I induced. I was referring to the constraints by HESS (and other IACTs) coming from observations of the Galactic Center, not from dwarfs. The constraints from the GC area may be relevant and apply to masses within the range considered in the paper. See for instance, for the historical evolution and the current status, and restricting only to HESS: arXiv:1103.3266, 1509.04123, 1607.08142, 2207.10471.
By mistake I pointed out the reference 2108.13646, which reports the constraints from HESS from dwarfs. These indeed are not competitive, in the current context. I apologize for that.
The authors have written:
-We have indeed not implemented the limits from charged cosmic rays as reported by various papers, as recent studies have shown that large uncertainties exist in cosmic ray propagation models, which have not previously been taken into account. These limits are also not particularly stringent, and will not change the main conclusions of the paper.
This response is not particularly satisfying. The authors should substantiate their statements more soundly.
First, to my knowledge uncertainties have been taken into account since a long time, and they have gone down significantly recently.
Second, what is the basis by which the authors claim that the limits are not stringent? They should have cited a reference or an estimate. For instance, the constraints based on AMS antiprotons (see fig 13, upper panel, of 1712.00002, the reference provided in the first report) may have an impact, if one compares with Fig. 5 of the present paper.

Author: Jochem Kip on 2022-11-15 [id 3017]
(in reply to Report 3 on 2022-11-07)Best,
We thank you for your report, we would of course like to comment on some of your points and address some concerns you have:
Regarding the discussion of the neutrino spectra. There appears to have been a compilation problem with the Feynman diagram, this has been fixed. The awkward-reading sentence has also been addressed.
The HESS limits have been implemented. Notably we do use the NFW limits, as opposed to the HESS Einasto limits, in order to provide a fair comparison. Seeing as the other indirect detection limits we implemented use the NFW DM profile. Including these limits does not change our results in any way.
We have provided references regarding the uncertainties in antiproton searches in the introduction of the text. We have additionally added text in section 4.4 detailing why we do not include the antiproton limits. But more importantly, most limits coming from antiproton DM searches are relevant in the 50-100 GeV DM mass range, which is not where our solutions lie. Even when included, no annihilation channels going to neutrinos will be excluded due to the extremely low antiproton yield of these channels.
Kind regards,
The authors

---

## Round 3 · List of Changes

-Added references and some text to the introduction
-Clarified the scanning procedure
-Clarified that point density is not indicative of statistical significance
-Added the tentative LZ direct detection limits
-Expanded upon the neutrino spectra analysis
-Some additional clarifications as requested by the referees in the various reports
-Clarified the scanning procedure
-Clarified that point density is not indicative of statistical significance
-Added the tentative LZ direct detection limits
-Expanded upon the neutrino spectra analysis
-Some additional clarifications as requested by the referees in the various reports

---

## Editorial Decision

published